# Using Algorithmic Transformations and Sensitivity Analysis to Unleash Approximations in CNNs at the Edge

**DOI:** 10.3390/mi13071143

**Published:** 2022-07-19

**Authors:** Flavio Ponzina, Giovanni Ansaloni, Miguel Peón-Quirós, David Atienza

**Affiliations:** Embedded Systems Laboratory (ESL), École Polytechnique Fédérale de Lausanne (EPFL), 1015 Lausanne, Switzerland; giovanni.ansaloni@epfl.ch (G.A.); miguel.peon@epfl.ch (M.P.-Q.); david.atienza@epfl.ch (D.A.)

**Keywords:** approximate computing, CNN quantization, ensembling methods

## Abstract

Previous studies have demonstrated that, up to a certain degree, Convolutional Neural Networks (CNNs) can tolerate arithmetic approximations. Nonetheless, perturbations must be applied judiciously, to constrain their impact on accuracy. This is a challenging task, since the implementation of inexact operators is often decided at design time, when the application and its robustness profile are unknown, posing the risk of over-constraining or over-provisioning the hardware. Bridging this gap, we propose a two-phase strategy. Our framework first optimizes the target CNN model, reducing the bitwidth of weights and activations and enhancing error resiliency, so that inexact operations can be performed as frequently as possible. Then, it selectively assigns CNN layers to exact or inexact hardware based on a sensitivity metric. Our results show that, within a 5% accuracy degradation, our methodology, including a highly inexact multiplier design, can reduce the cost of MAC operations in CNN inference up to 83.6% compared to state-of-the-art optimized exact implementations.

## 1. Introduction

The edge computing paradigm [1] is fostering a revolution in Artificial Intelligence (AI), impacting scenarios ranging from personalized healthcare to autonomous driving and automatic text generation [2,3,4]. By shifting data processing from the cloud to end devices, edge computing enables increasing efficiency dramatically, because data acquisitions do not need to be transmitted over energy-hungry radio links. Moreover, local processing results in low latencies and high responsiveness, which are often crucial for edge devices.

Edge AI applications are often realized as Convolutional Neural Networks (CNNs). These architectural models are usually structured as a sequence of processing layers, each extracting increasingly abstract features from the input data to perform classification or detection tasks. Recent research efforts [5,6] have highlighted that the architectural redundancy of CNNs makes these models resilient to perturbations. To increase their robustness even more, the authors of [7] observe that algorithmic optimizations can increase the intrinsic resiliency of CNNs against errors. They propose a solution that transforms a target single-instance CNN into a resource-constrained ensemble of CNNs, improving robustness towards memory upsets while not increasing computational workload and memory requirements.

Nevertheless, CNN inference often demands the execution of millions of multiply–accumulate (MAC) operations and large memories to store parameters, straining the capabilities of ultra-low-power embedded systems. Two main optimization avenues have been proposed in the literature to address this challenge. Pruning approaches [8] entail the removal of specific neural connections or entire computational blocks from CNN models, while quantization strategies [9] reduce the bitwidth of CNN weights and/or activations.

A third path to optimize the run-time performance of CNNs is the use of approximate operators that trade arithmetic correctness for efficiency [10]. In this work, we introduce a two-stage methodology where we employ inexact arithmetic in carefully selected CNN layers to further increase inference efficiency of CNN models. In contrast to previous works [11], a key aspect of our optimization loop is that accuracy degradation is effectively controlled, independently of the approximation degree of the multiplier itself.

Unfortunately, the impact of inexact circuits on CNNs output quality degradation cannot be evaluated at design time, when these operators are selected, because the impact on accuracy also depends on the model structure and the task complexity. The authors of [11] analyzed the impact of different inexact multipliers on the convolutional and fully connected layers of the VGG16 model, and found that the first and last layers are particularly sensitive to approximation. Therefore, to obtain a positive accuracy vs. efficiency trade-off, they suggest a hybrid approach where only the central layers are executed using approximate multipliers. In this work, we demonstrate that inexact multipliers have a limited impact on efficiency when applied alone to baseline CNN models, especially when compared to quantization. Nevertheless, a judicious use of these circuits in highly optimized (quantized) models can further improve efficiency. Hence, we carefully map them to execute specific CNN layers and combine them with orthogonal state-of-the-art CNN optimization strategies to fully exploit their benefits. To guide the selection of CNN layers where inexact arithmetic can be applied, while abiding by a certain user-defined accuracy level, we propose a heuristic method that evaluates the resiliency of individual layers by performing a sensitivity analysis. This approach logically separates the approximation degree of the employed inexact multiplier from the user-defined accuracy threshold, making these two values independent input parameters in our proposed methodology.

The contribution of this paper is three-fold:We demonstrate that, when applied to baseline CNN models, approximate multipliers can only marginally improve inference efficiency while preserving accuracy. Thus, we combine inexact computing with other optimization strategies, showing how approximate multipliers can be effectively employed to fully exploit energy savings.We present a two-stage accuracy-driven methodology that combines ensemble methods, heterogeneous quantization, and a selective use of inexact operators to improve energy efficiency in CNNs at the edge, while increasing their resilience towards the noise introduced by approximate multipliers.To introduce the use of inexact arithmetic in our optimized model, we propose a novel heuristic-based approach that exploits the results of a preliminary analysis evaluating the sensitivity to approximation of individual CNN layers; thus, it ultimately tailors the design to user-specified accuracy requirements, irrespective of the approximation level of the selected multiplier.

The rest of the paper is organized as follows: in Section 2, we place our work in perspective of related research efforts. Then, the proposed methodology is detailed in Section 3. The adopted experimental setup is presented in Section 4, while results are discussed in Section 5. We summarize our findings in Section 6.

## 2. Related Works

### 2.1. Quantization

While CNNs are typically trained using floating-point representations for intermediate values (activations) and parameters (weights), it is known that, during inference, more energy- and storage-efficient alternatives can be adopted with little impact on accuracy. As an example, Reagen et al. [12] propose an implementation where either 8 or 16 bits are employed to represent weights, while Courbariaux et al. [13] introduce binarized CNNs, in which both weights and activations are constrained to be either −1 or +1, thus using a single bit for their representation, but at the cost of important accuracy degradation.

Recent works propose heterogeneous per-layer quantization strategies in which the activation and weight bitwidths are assigned according to the layer robustness, to increase efficiency while preserving accuracy. While, in principle, quantization may be performed considering arbitrary bitwidths [14], such fine-grained flexibility usually incurs vast overheads. Additional logic can instead be minimized when the adopted quantization levels are SIMD standard bitwidths (e.g., quantization on 16, 8, or 4 bits, as in [12,15]), since, in this case, word-level parallelism can be effectively employed.

### 2.2. Ensembles of CNNS

Ensemble methods targeting CNNs have been investigated and proved to improve classification accuracy, at the cost of dramatically increasing memory and computational requirements due to the replication and deployment of several CNN models [16]. To avoid this pitfall, the authors of [7] compress CNNs via filter pruning by a factor equal to the number of instances deployed in the ensemble, so that the resulting architecture does not require more computation and storage than the single-instance original model. Their proposed resource-constrained ensembles are more accurate and robust against memory errors than equivalent single-instance CNNs. In this work, we consider resource-constrained ensembles of CNNs in a different context: as an avenue to increase CNNs’ tolerance towards arithmetic approximations.

### 2.3. Approximate Computing in CNNS

In a broad sense, the approximate computing paradigm encompasses strategies trading off the exactness of computed results with computing performance metrics such as run-time and/or energy [17]. In the context of this paper, methods related to Approximate Logic Synthesis are of particular relevance. In particular, they are able to derive inexact, but extremely energy efficient, arithmetic circuits for commonly used operators [18]. These operators can then be employed as building blocks for complex accelerators [19].

This approach is of particular interest when targeting CNN accelerators, as they usually present highly parallel and computation-intensive structures, where a major contribution to resource and energy budgets is the arithmetic logic in their datapaths [20,21]. Indeed, several studies have advocated the use of inexact circuits in CNNs [11,22]. The authors of these works highlight that, when considering CNNs, multipliers are the most amenable target for approximation. In particular, multipliers typically present a high energy footprint (e.g., with respect to adders) and because neural networks require a very high number of multiplications. We also focus on approximating multiply operations in our work, but, as opposed to [11,22], we adopt an application mapping perspective, aiming to leverage the available energy-saving opportunities in inexact hardware target while controlling degradations in accuracy.

## 3. Proposed Methodology

To effectively explore the large space of candidate designs due to the combination of ensembling methods, heterogeneous quantization, and inexact operators to improve CNN inference efficiency, we employ the methodology summarized in Figure 1. Our methodology accepts as input a single-instance CNN. First, it applies the concept of embedded ensembles to increase the robustness of the CNN model. Second, it applies heterogeneous quantization to reduce the use of memory bandwidth and computational resources in each layer. Third, it analyzes the layers of the baseline CNN model to evaluate their sensitivity to approximation. Finally, it maps the obtained ensemble on approximate hardware resources, leveraging their lower power consumption to further increase inference efficiency. These four steps are implemented offline in two stages.

### 3.1. Stage A: Robustness-Aware CNN Optimization

Starting from a single-instance floating-point model, our optimization framework first derives the structure of an ensemble implementation that improves accuracy and robustness against data perturbations (I in Figure 1). As in [7], we use pruning and replication to build ensembles with no memory or computational overheads compared to the initial single-instance CNN. Specifically, to build an ensemble composed of *M* instances, we drop a certain number of convolutional filters from the initial CNN structure (i.e., coarse-grain filter pruning), until its memory and computational requirements are reduced by a factor *M*. The resulting pruned architecture is then replicated *M* times, and each instance is independently trained, starting from different (random) weight values, to increase variability and ultimately improve accuracy. Uniform quantization on 8 bits for the weights and 16 bits for the activations (i.e., 8/16 quantization) is applied during the last training epochs, without affecting the accuracy of the baseline floating-point model [12].

The generated ensemble of CNNs is then optimized by including a (further) heterogeneous quantization in the CNN instances building the ensemble (II in Figure 1). We consider each instance individually and proceed per layer in topological order, reducing the bitwidth of the operands to only 4 bits for the weights and 8 bits for the activations (i.e., 4/8 quantization). The 4/8 quantization level is applied to a certain layer if the resulting accuracy meets the user-defined constraint. Otherwise, the previous 8/16 quantization is retained. This process ends when all layers have been evaluated. CNN ensembling and per-layer quantization are employed synergically. On one side, the higher accuracy and robustness of ensembles serves as a support for unleashing more aggressive approximations. On the other side, a per-layer quantization reduces memory and computational requirements and improves efficiency by enabling the use of simpler (and therefore more efficient) multipliers for the execution of 4/8 quantized layers.

### 3.2. Stage B: Mapping on Inexact HW Resources

The use of approximate multipliers mandates a cautious approach, because relying entirely on inexact arithmetic can have a critical impact on accuracy. Thus, in the second stage of our proposed methodology, we adopt an accuracy-driven heuristic method to select, among the layers that are quantized at an 8/16 level, the ones robust enough to be implemented using the target approximate multiplier. The heuristic orders the layers according to their sensitivity (III in Figure 1). We measure sensitivity by instantiating the selected approximate multiplier in only one layer of the single-instance uniformly quantized model at a time and evaluating the resulting inference accuracy. This analysis is performed on the initial single-instance model. We extend the obtained results to the CNN instances composing the generated ensemble. On one side, we have observed that the same analysis, performed on each CNN instance, produced very similar results. Hence, these results suggest that layers’ resiliency may be more closely associated with their size and structure than with their actual weight values. On the other side, such an approach reduces optimization run-time as the analysis is executed only once. Moreover, in contrast to an impractical exhaustive exploration, this heuristic can efficiently scale to large CNN applications. Indeed, being *L* the number of convolutional and fully connected layers, the computational complexity of our strategy is O(L). Indeed, the per-layer quantization and use of inexact multipliers are applied in sequence, and are themselves of linear complexity. In contrast, an exhaustive search, while it would guarantee to find optimal solutions, would also need to enumerate all the possible configurations. Hence, its complexity is $O(3^{L})$, since 3 alternative implementations exist for each layer (i.e., 4/8 quantization, 8/16 quantization, 8/16 quantization with inexact multipliers).

As we show in Section 5, reducing the bitwidth of the operands involved in MAC operations has a larger impact on energy consumption than the use of approximate logic (on a larger bitwidth). For this reason, our methodology applies heterogeneous quantization before introducing approximate operators. Additionally, we observe that using any approximate multipliers on 4/8 quantized layers has an adverse impact on accuracy. Therefore, approximation is possibly applied only to those layers that, after the heterogeneous quantization are still kept at an 8/16 bitwidth.

Finally (IV in Figure 1), we combine the results of the sensitivity analysis with the optimized ensemble, iteratively introducing approximate multipliers in the CNN instances, starting from the least sensitive layers. This phase terminates when no further layer can be approximated while abiding by the accuracy constraint.

## 4. Experimental Setup

To gauge the potential benefits of our strategy, we consider in this work a diverse collection of CNN applications, comprising AlexNet [23], VGG16 [24], GoogLeNet [25], ResNext [26] and MobileNet [27]. In all cases, we adopt Top-1 as accuracy metric, and CIFAR-100 as dataset [28]. All the benchmarks are trained in PyTorch [29], using fake quantization functions as in [30] for the last 20 training epochs. As in [7], we build ensembles containing 2, 4, or 8 instances and present in our results the configuration achieving the highest accuracy. Across experiments, efficiency is measured as the energy required by all exact and inexact multiplications executed in an inference. The energy impact of MAC operations at the chip level is architecture-dependent: it may be relatively low in single-core platforms where data movements account for the largest fraction of energy consumption, but it can dominate in multi-core edge AI accelerators comprising hundreds of processing elements [31].

Compared to the approximation approach implemented in [6], where inexactness is achieved reducing operands’ precision (i.e., similar to what quantization does), we simulate the behavior of the employed (possibly inexact) multipliers in a C++ inference solver that also measures inference accuracy. Additionally, in contrast to [11], where approximation is applied to float16 arithmetic by using approximation matrixes that simulate inexact operators, we consider two different integer approximate multipliers, as presented in [32]. Their structure is derived by employing a multi-objective Cartesian genetic programming approach (CGP), while using different exact implementations as starting point. Among the large number of potential candidates provided by this library, we select two inexact multipliers that vastly differ in the magnitude of introduced arithmetic approximation, to showcase the effect of operators with either a large or a small degree of inexactness. We adapt their structure to match the bitwidth of input and output operands in our quantized layers: for example, the 16-bit multipliers in [32] are overdimensioned for 8/16 layers, as one input operand (the weight) requires only 8 bits. Therefore, we modify the original Verilog implementation, adjusting the bitwidth of input and output operands, as well as the bitwidth of the connected internal components. We characterize the power consumption of the circuits using Synopsys Design Compiler, employing HVT cells from the 40LP TSMC technology library (40 nm, low power). The error induced by approximation is measured in terms of Mean Relative Error (MRE) by running a simulation over all the possible input combinations. The synthesis and simulation results are summarized in Table 1. Exact16 and Exact8 are exact multipliers used in 8/16 and 4/8 layers, respectively, while MulF6B and Mul8VH are approximate multipliers used only in 8/16 layers, with the latter offering more energy savings at the cost of a larger impact on precision. In contrast to [22], where the use of inexact multipliers is limited to fully connected layers, we also introduce approximation in convolutional ones, because they account for a large percentage of MACs in our benchmarks. As a proof of concept of our approach, we consider a target system featuring two exact multiplier implementations (Exact16 and Exact8) and an approximate one (either MulF6B or Mul8VH in our experiments).

## 5. Experimental Results

### 5.1. Synergic Use of Ensembles and Heterogeneous Quantization

As reported in Table 1, the energy savings achieved through arithmetic approximation alone are of 15% when using the MulF6B inexact multiplier. Higher savings can be obtained by considering more aggressive implementations such as Mul8VH that also introduce larger perturbations. Alternatively, 4/8 bits quantization reduces the energy consumption of multiply instructions by 85%, even when using an exact multiplier (Exact8). This finding motivates our iterative approach, where quantization is applied before the introduction of inexact multipliers, hence enabling a larger number of layers to be executed using Exact8 (II in Figure 1). We present in Figure 2 the accuracy/efficiency trade-off achieved in different benchmarks and design configurations. Black circles correspond to the baseline implementation used as a reference for comparison and refer to single-instance implementations adopting the same 8/16 bitwidth in all layers. In line with what has been already described in [7], we observe that ensemble-based implementations (blue circles) improve the accuracy of single-instance CNNs, because the limited accuracy drop of individual pruned CNNs is largely compensated by the higher generalization capability of ensembles. Green circles represent the second step of our proposed methodology and correspond to the heterogeneously quantized ensembles. Layers quantized to 4/8 bits are computed using the Exact8 multiplier, which produces energy savings ranging from 22.1% in ResNext up to more than 80% in VGG16.

As previously suggested, it is possible to further increase the energy savings by substituting the Exact16 implementation used in 8/16 layers with an inexact alternative, such as MulF6B (★) or Mul8VH (▪). Instead, we keep the Exact8 multiplier in 4/8 layers, since arithmetic approximations in small-bitwidth operators do not result in high overall energy gains, but have a large impact on accuracy.

On one hand, Figure 2 shows that approximate multipliers can be effectively employed together with heterogeneous quantization to obtain up to 39.9% additional energy savings. On the other hand, the approximation introduced by certain multipliers when indiscriminately used in all 8/16 layers can be too large, resulting in an unacceptable accuracy degradation. This observation motivates a more judicious selection of the layers where inexact multipliers should be used, limiting their impact on accuracy (III in Figure 1).

### 5.2. Sensitivity Analysis for a Layer-Based Selective Use of Inexact Arithmetic

We observed in Figure 2 that highly inexact multipliers, such as Mul8VH, can produce high energy savings, but at the cost of a critical impact on classification accuracy. Hence, in this section, we investigate the robustness of individual layers against the arithmetic approximations induced by an inexact multiplier, to select those robust enough to endure approximate computation. To do so, we employ such an inexact operator in one layer at a time, using the Exact16 implementation to execute all the other layers. We use the obtained inference accuracy as a metric to determine each layer’s resiliency. We consider the Mul8VH multiplier to describe our analysis because its large impact on accuracy better illustrates our results.

The outcome of this analysis is summarized in Figure 3, where we show the accuracy drop corresponding to the use of Mul8VH in different layers. This analysis indicates that the robustness of convolutional and fully connected layers varies in different benchmarks. As an example, in AlexNet, the intermediate layers are the most robust ones (i.e., they cause the least accuracy drop when approximated), while the most robust layers of VGG16 and GoogLeNet are the last ones. Similarly, no clear relationship exists between the robustness of a certain layer and the number/percentage of MAC operations required for its execution.

In all the evaluated benchmarks, the first convolutional layer is always highly sensitive to arithmetic approximation. Since it also executes a small fraction of the total MAC operations, a naive approach that simply performs inexact arithmetic in the entire model except for the first layer decreases the accuracy degradation by 6.9%, while reducing the potential energy savings by just 1.5% on average. Nevertheless, a more accurate approach can lead to a better accuracy/efficiency trade-off. Consequently, we include the described sensitivity analysis in our methodology (III in Figure 1), and use it to order the layers from the least to the most sensitive ones. Next, at each iteration, the target approximate multiplier is iteratively employed in one additional 8/16 quantized layer of each CNN instance forming the ensemble, until the accuracy degradation becomes unacceptable. As opposed to our solution, an exhaustive exploration to select the optimal mapping of inexact multipliers in the described CNN design is an impractical approach. Indeed, even for simple architectures composed of relatively few layers such as AlexNet, an exhaustive search would take more than two months to complete when run on a Tesla V100 GPU (from NVIDIA, Santa Clara, CA, USA), while our heuristic approach terminates in just a few hours.

### 5.3. Overall Methodology Evaluation

In the previous sections, we have described individual steps of the methodology illustrated in Section 3. Instead, in this last round of experiments, we evaluate the optimized design at the output of our methodology, in terms of accuracy and energy reduction of multiply operations. To this end, we consider heterogeneously quantized ensembles of CNNs and employ Mul8VH as a candidate approximate multiplier. Then, we iteratively select layers to be approximated as dictated by the sensitivity metric. We have previously shown in Figure 2 that the use of the highly inexact Mul8VH in every 8/16 quantized layer incurs very high accuracy degradations. We herein showcase that, by instead selectively employing it only in robust layers, high energy gains can be achieved while preserving accuracy. To include a highly optimized baseline in these experiments, we compare the final outcome of our methodology with the presented heterogeneously quantized ensemble using exact arithmetic operators. Additionally, we also compare our ultimate solution with the same quantized ensemble using the target approximate multiplier (i.e., Mul8VH) in all 8/16 layers.

The results for a maximum accuracy drop of 5% are shown in Figure 4. Red squares indicate the achieved accuracy/energy trade-off of our solution. Conversely, the green circles on the left of each series report the energy/accuracy of the described baseline, while the rightmost black squares represent the solution relying on inexact arithmetic only in 8/16 layers. This comparison showcases that our proposed methodology outperforms state-of-the-art alternatives for a certain user-defined accuracy level, further increasing energy savings up to 21% compared to heterogeneous quantization alone and harnessing up to 78% of the energy gains achievable when employing Mul8VH in all CNN layers (a solution that results in unacceptable accuracy degradations). The limited energy gains obtained in VGG16 when introducing inexact multipliers (i.e., less than 4%, even when relying on inexact arithmetic in all 8/16 layers) are due to the high effectiveness of the aggressive heterogeneous quantization. Figure 2 shows that VGG16 achieves almost 80% of energy reduction via quantization (green circle). This result indicates that the majority of its layers employs a 4/8 quantization and can therefore use the Exact8 multiplier. As a consequence, the limited number of layers kept to an 8/16 quantization level prevents further significant energy reductions. When compared to baseline exact single-instance implementations, our results achieve 59.4%, 83.6%, 42.7%, 39.9%, 82.6% energy reductions for an accuracy degradation limited to 5% in AlexNet, VGG16, GoogLeNet, ResNext and MobileNet, respectively.

To further demonstrate the benefits of using a proper selection of approximated layers, we present a detailed exploration targeting the VGG16 benchmark in Figure 5. Therein, we show the accuracy obtained by single-instance CNNs and ensembles, where the Mul8VH multiplier is employed in an increasing number of layers, comparing the achieved accuracy when using our sensitivity-based approach with an alternative in which layers are approximated in topological order. Our results confirm the additional robustness of ensembles and demonstrate that the topological approach fares far worse than our proposed sensitivity-based one, because fewer layers can be arithmetically approximated for a target accuracy or, alternatively, far lower accuracy is obtained for the same number of approximated layers. Indeed, Figure 5 indicates that, with our approach, Mul8VH can be used in 7 layers in VGG16 and in 13 layers in the corresponding ensemble, while still limiting the accuracy degradation to 5% in each version, and achieving 19.6% and 43.7% energy reductions, respectively. In contrast, introducing approximation following a topological order limits the achievable energy reduction to 10.2% and 31.9% for the same accuracy level.

### 5.4. Area Impact of Deploying Multiple Multiplier Circuits

To support the execution of both exact and inexact arithmetic in 8/16 layers (i.e., using either Exact16 or an approximate multiplier), and the exact arithmetic in 4/8 layers (i.e., using the Exact8 implementation), three different multipliers have to be deployed. As an example, considering MulF6B as a candidate approximate multiplier, a total area of 1159 μm^2^ is required to instantiate the two exact multipliers, Exact16 and Exact8, and the selected inexact multiplier, MulF6B. The resulting configuration has an area overhead of 86% with respect to Exact16 alone. Nonetheless, the Exact16 multiplier can be implemented by combining Exact8 units, resulting in just 6% area increment. This solution enables the execution of two 8-bit multiplications simultaneously (i.e., SIMD) when the multiplier is used in 4/8 layers, which can be exploited at the application level to speed-up inference execution. Results are summarized in Figure 6 and show that the area overhead for executing both exact and inexact arithmetic in our design ranges from 31%, when considering the highly inexact Mul8VH, up to 71% for the MulF6B. Consequently, the ability of our methodology to handle highly inexact multipliers can limit the area overhead. Indeed, their deployment alongside the exact multiplier in the final design demands for a lower area footprint with respect to less inexact implementations. At the same time, their use still increases efficiency and guarantees a user-defined output quality thanks to a judicious per-layer mapping. Finally, although the trade-off between accuracy and efficiency could be explored more deeply instantiating different approximate multipliers in different layers (i.e., according to their degree of resiliency), our results indicate that the significant area overhead of these circuits may limit such an approach.

## 6. Conclusions

In this paper, we have proposed a new methodology to design highly efficient CNNs by concurrently exploiting ensembling, heterogeneous quantization, and inexact multipliers to reduce inference energy while controlling accuracy degradation. Our results indicate that ensembling is able to improve both accuracy and robustness against arithmetic approximation, without any overhead in terms of computational and storage resources. On a diverse and representative collection of CNN benchmarks, we have shown in this work that, thanks to a sensitivity-based analysis, approximate multipliers can be effectively employed in conjunction with heterogeneous quantization, enabling energy savings in multiply operations up to 83.6%, within a 5% limited loss of accuracy reduction. Our methodology favorably compares with homogeneous alternatives, as employing highly inexact multipliers in the homogeneous case results in very high performance degradations. Moreover, we have shown that lower benefits (limited to 15% in case of MulF6B in our experiments) can be harnessed by homogeneously employing multipliers with a low approximation degree.

## Figures and Tables

**Figure 1 micromachines-13-01143-f001:**
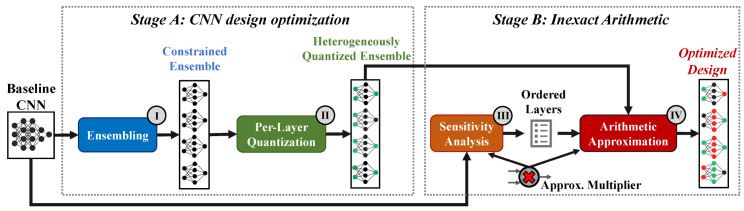
The proposed two-stage methodology to CNN inference efficiency. It combines the use of heterogeneously quantized ensembles with a selective use of inexact arithmetic operators.

**Figure 2 micromachines-13-01143-f002:**
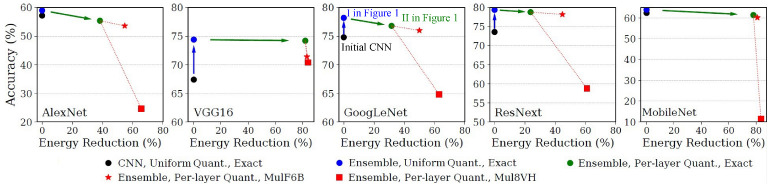
Accuracy/efficiency trade-off in single-instance CNNs (black) compared to uniformly (blue), and heterogeneously (green) quantized ensembles. The use of inexact multipliers in all the layers kept at an 8/16 quantization level (red markers) results in large accuracy losses.

**Figure 3 micromachines-13-01143-f003:**
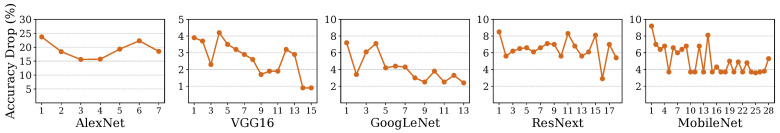
Accuracy drop when using Mul8VH in a single selected layer. The X-axis indicates the layer index in which Mul8VH is adopted.

**Figure 4 micromachines-13-01143-f004:**
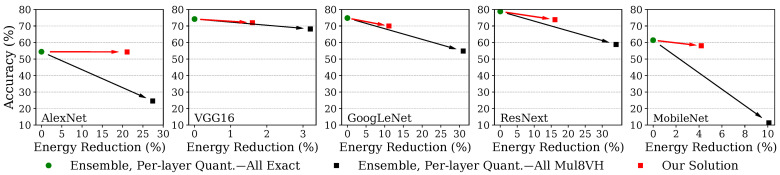
Our solution, where approximate arithmetic is executed only in specifically selected layers (red squares), is compared with heterogeneously quantized ensembles either employing Exact16 (green circles) or Mul8VH (black squares) in all 8/16 layers. The overall energy savings achieved only via quantization in the presented ensembles (green circles, here considered as baseline implementations) can be retrieved from Figure 2 (also marked as green circles).

**Figure 5 micromachines-13-01143-f005:**
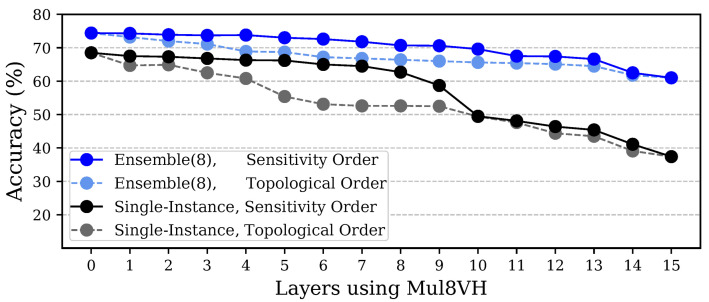
Accuracy drop when using Mul8VH in the *N* least sensitive layers (solid lines) or in the first *N* topologically ordered layers (dashed lines) of VGG16.

**Figure 6 micromachines-13-01143-f006:**
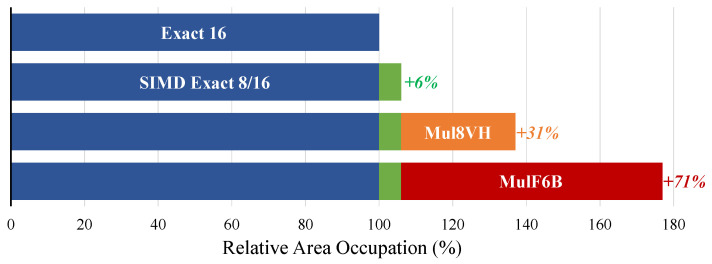
Comparison of the area of different multiplier combinations. Our proposal uses the exact multiplier for 4/8 layers in a SIMD configuration, while, in 8/16 layers, it employs either an inexact multiplier or the exact one in its native 16-bits configuration. By enabling the use of highly inexact multipliers (e.g., Mul8VH), our methodology reduces area overheads.

**Table 1 micromachines-13-01143-t001:** Operands bitwidth, mean relative error and power characterization of the multipliers used in our experiments.

	Bitwidth (IN1 × IN2)	MRE (%)	Power (μW)	Area (μm^2^)
Exact16	(8 × 16)	N/A	277.5	622.5
MulF6B	(8 × 16)	5.9 × 10^−5^	237.3	441.7
Mul8VH	(8 × 16)	1.9 × 10^−3^	137.3	192.9
Exact8	(4 × 8)	N/A	39.9	94.8

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
