# Peer review of "Using Algorithmic Transformations and Sensitivity Analysis to Unleash Approximations in CNNs at the Edge"

_micromachines, 2022, doi:10.3390/mi13071143_

Round 1
Reviewer 1 Report
There is one common understanding in ML model study: improvements and invocations need to be done on SOTA models and adequate datasets in the field. Otherwise, garbage input will probably lead to garbage output.
This paper did not fulfill either of the requirements, which means first, it is not using an adequate dataset for the problem, for the image classification problem here, the Imagenet dataset should be the baseline, not cifar-100; second, even we agree cifar-100 is good enough for the problem here, none of the ML models chosen in the paper are SOTA models. With accuracy < 80%, there are all ranking >100 in the benchmark list: https://paperswithcode.com/sota/image-classification-on-cifar-100.
Due to the reasons mentioned, the reviewer could not judge the work because the experiment is not done in the right way.
Reviewer 2 Report
The paper entitled "Using Algorithmic Transformations and Sensitivity Analysis to Unleash Approximations in CNNs at the Edge" presents an interesting design methodology for CNNs, by ensembling, heterogeneous quantization, and approximate multipliers targeting lowering CNN inference energy while maintaining accuracy. To the reviewer, the work is solid, theoretically sound, and practically useful. Here I listed my main observations on this work:
1- The paper gives a nice introduction to quantization in CNNs and a decent background review on approximate computing in CNNs.
2- Both methodology stages, i.e., robustness-aware CNN and optimization and mapping on inexact HW resources are technically sound and explained well, however, the inexact multipliers structure used in stage 2 could be explained more clearly. The ability of the presented methodology to handle highly inexact multipliers is interesting.
3- The evaluation methodology and results are solid and comprehensive and show the pros and cons of the proposed method.
4- From a presentation perspective, the paper is well-structured from section to section and overall is nice.
Reviewer 3 Report
To design highly efficient and robust CNNs from a single-instance baseline CNN for tolerating arithmetic approximations, a novel framework with an algorithm of sensitivity analysis is proposed in this article.
A series of experiments show that a constrained ensemble method, an algorithm of sensitivity analysis and approximate multipliers can be effectively employed in conjunction with heterogeneous quantization, achieving various degrees of energy savings in some state-of-the-art CNNs with a controlling accuracy degradation.
This article is original and has a practical idea, but some areas need to be improved as follows:
Major Issues:
1. Line 165 “Finally, we combine the results of the sensitivity analysis with the optimized ensemble...” Why use the sensitivity analysis of Baseline CNN? Suppose it has a very similar sensitivity layer order with the optimized ensemble. In that case, you should prove/display the relationship since Line 144 - Line 145 (The resulting pruned architecture is … starting from different weight values) mentioned that the optimized ensemble had trained from random weight values. Furthermore, why not make a sensitivity analysis on the optimized ensemble?
2. Figure 2 includes the accuracy/efficiency trade-off in Excat16, Mul8VH and MulF6H. However, Figure 3 only compares the accuracy/efficiency trade-off in Excat16, MulF6H and your solution, but not Mul8VH. Why is the result of Mul8VH missing in Figure 3?
3. Figure 6 exhibits that the relative area occupation will increase if using inexact multipliers (Mul8VH and MulF6H). But Table 1 demonstrates that the area of Mul8VH and MulF6H is smaller than Exact16.
Minor Issues:
1. Line 143 “…until its memory and computational requirement are reduced by a factor M.” If the “M” does not present the same content/value with the other two “M” which be mentioned in Lines141(“M instances”) and Lines 144(“M times”), the “M” should be replaced by another alphabet or attached explanation in case cause ambiguity.
2. The legend of Figure 2 is very unclear; some of the symbols are missing.
3. You did not reference Figure 6 in section 5.4 (Line337 – Line 355)
Round 2
Reviewer 1 Report
The reviewer has read through the author's reply. However, the reviewer could not agree with the response for the following reasons:
1. You could not use the same standard 4 years ago to judge the publication now in the ML area.
2. For the paper "Ares: A framework for quantifying the resilience of deep neural networks” ACM/IEEE DAC, 2018" listed in your response, they did use ImageNet to evaluate their VGG16 and ResNet50 models and again, it is published in 2018. Please check Table 1 in their paper, authors should not ignore this.
3. For the other two papers, their contributions are on the hardware design, and they reported much more detailed results on the hardware, instead of pure accuracy. If the author wants to compare them, similar quality of hardware results should be reported and show improvements.
4. In the end, if authors want to evaluate the impact of approximated hardware for NN on the edge, the author should use SoTA edge NN models, like those for mobile phones. There are a lot of models, and they use very different methods to compress the NN size, such as depth-wise convolution in MobileNetV2, V3, and other models. The impact of approximation will probably be different on depth-wise convolution to dense convolution or fully connected layers.
Reviewer 3 Report
The authors have addressed all the issues and the manuscript can be accepted now.